# POET: Prompt-offset Tuning for Continual Few-shot Action Recognition

## Abstract

As virtual reality and augmented reality is redefining how users interact with computing devices, research in action and gesture recognition is indeed gaining prominence. Typically, these models deployed on AR/VR devices are trained in their factory, with large proprietary datasets. Though this training would cover the major set of activity and gestures classes, the user should ideally be able to add newer classes to the model, without forgetting the base set of classes. Importantly, the user would be able to provide only few samples per class in this process. In-order to protect the user's privacy, the setting should also not allow storage and replay of a data sample, for future learning. We formalize this pragmatic problem setting as *privacy aware few-shot class incremental learning for activity and gestures*. Towards this end, we propose a novel strategy, *POET: Prompt-offset Tuning*. Unlike other prompt tuning approaches that demand access to transformer models pretrained on a large amount of data, our approach demonstrates the efficacy of prompting on a significantly smaller model trained exclusively on the data from the base classes. Additionally, we take advantage of the temporal sequencing in the data stream of actions and gestures to propose a unique temporal-ordered learnable prompt selection and prompt attachment. To evaluate our newly proposed problem setting, we introduce new benchmarks on NTU RGB+D dataset for action recognition and SHREC-2017 dataset for hand gesture recognition.

## 1 Introduction

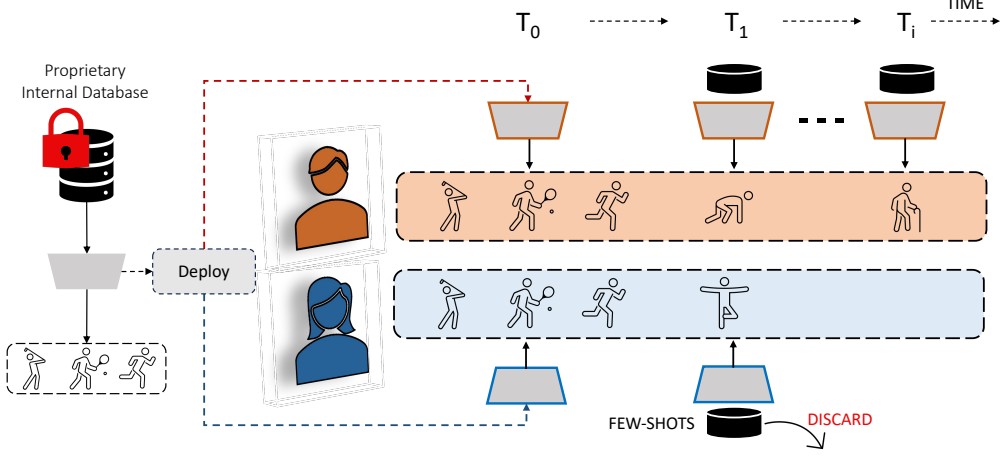

Figure 1: Each user's device initially recognizes the same set of base action classes ($T_0$). Each user can then separately add their own (new) action classes whenever they want (at time $T_1, ..., T_i$) to their respective model, by supplying just a few training examples (few-shots) of each new class. This learning paradigm relies only on the continual model for action recognition of all classes seen till that time stamp, discarding user-sensitive data as soon as the model is updated on few-shots.

Virtual, augmented, mixed and extended reality devices today have increasingly accurate pose estimation that tracks body and hand pose over time, which can be used to recognize human activity and hand gestures. Developers can enable gestures as input for users to interact with virtual environments with their hands. Human-robot interactions can be facilitated by activity recognition, such as a digital assistant reacting to the user's actions. Although devices can come with a model that recognizes a predefined set of actions and gestures, developers and users may want to expand or customize the experiences and functionality of the devices by adding and defining their own actions and gestures. To easily and securely add new classes we need a continual learning approach that (1) requires a minimal amount of training data so developers and users can added new classes by collecting just a few examples and (2) does not need access to previous data and is thus privacy preserving. In this work, we introduce the concept of privacy-preserving few-shot class incremental action recognition. As depicted in Figure 1, we expand the capabilities of the device's default (base) machine learning action recognition model, empowering users to introduce new action classes that were previously unknown to the model. Importantly, we provide the assurance that once the model is updated with these new classes, all associated data is completely discarded.

Addressing the demand for continual learning among users often involves a straightforward yet impractical approach: fine-tuning the default model, which is initially trained on proprietary company data, each time a user requires new action classes for their application. This process proves to be both costly and time-consuming. In contrast, prior works like MetaSense Gong et al. (2019) have explored adapting the base model to users' specific conditions through a few-shot meta-learning approach. However, the resulting adapted model is limited to working only with the new classes, rendering it non-functional for the default base model classes. Consequently, we are confronted with a fundamental challenge known as few-shot class-incremental learning (FSCIL). Our goal is to tackle this dynamic learning paradigm in such a way that the model can adapt to new knowledge without erasing existing knowledge, i.e., catastrophic forgetting (McCloskey & Cohen, 1989). In terms of input modality, we have devised our solution around 3D skeleton temporal sequences. These sequences possess several advantages, including invariance to lighting conditions, viewpoints, and environments, all while preserving user privacy.

Our solution is to construct a single model with a dynamically expanding classifier, plastic enough to learn new user classes with high accuracy while maintaining stability to retain performance on all the classes seen till now (stability-plasticity dilemma (Abraham & Robins, 2005)). A cost efficient and elegant solution to this problem is to explore learnable input-conditioned control vectors being trained along with the classifier, while keeping the backbone fixed. Inspired by the existing works in prompt tuning and visual continual prompting, we explore prompt tuning for few-shot continual action recognition and find interesting revelations.

We should highlight three fundamental distinctions that set us apart from all existing studies on continual prompting: (i) First, existing continual visual prompting techniques typically assume access to a 'generalist' transformer backbone that has undergone extensive pre-training. These backbones, such as ViT-B/16 pretrained on ImageNet21K, as demonstrated in Wang et al. (2022c); Smith et al. (2023), or a pretrained CLIP model as seen in Wang et al. (2022a); Villa et al. (2023), form the foundation of their approaches. Studying continual learning on ImageNet-R with a feature extractor backbone *pre-trained* on ImageNet21K can be thought of as learning distribution shifts on top of pre-existent knowledge. In this regard, L2P Wang et al. (2022c) cites the assumption of large pre-trained transformer backbones as a limitation of their work, APT Bowman et al. (2023) analysis show that quality of pre-training is pertinent to their prompt performance and very recently, Tang et al. (2023) proves that SOTA continual visual prompting works fail when there is a large semantic gap between pre-trained and continual tasks. Motivated by the lack of existing large pre-trained models or datasets for skeleton-based action recognition, we foray into exploring if prompting yields merit by itself - if we remove the backbone all existing works attribute the credibility of their prompt tuning to. (ii) Secondly, we recognize unique challenges and insights arising from *few-shot* class incremental prompting. This perspective sets us apart from existing works, which are primarily tailored for standard (fully supervised) class-incremental learning (CIL). (iii) Finally, in a first, we explore prompting of *non-transformer* based Graph Neural Networks (GNNs) for *temporal* skeleton action recognition data.

## 2 RELATED WORK

**Action Recognition and Continual Learning.** In recent years, *skeleton-based* action recognition has been gaining prominence as the preferred modality in applications that require low-shot action recognition capabilities such as medical action recognition (Ma et al., 2022; Zhu et al., 2023). This is because skeletons offer a robust and compact alternative to videos which are ill-suited to low-shot regimes due to their high dimensionality and variance under background conditions. Hence, we distinguish ourselves from all existing video continual learning works Park et al. (2021); Villa et al. (2022; 2023). Authors in Li et al. (2021a) first attempted 3D skeleton-based (standard) class-incremental action recognition using dynamic network expansion and exemplar replay. More recently, BOAT-MI Aich et al. (2023) proposes a data-free solution to class-incremental hand gesture recognition, learning a single-class in each new task.

**Few-Shot Class-Incremental Learning.** As compared to standard CIL, FSCIL is a more rational continual learning setting which has been widely studied due to its unique dual-challenge of overfitting to novel class few-shots and the heightened (often complete) forgetting of old knowledge as soon as the base model is fine-tuned on future few-shot data Tao et al. (2020); Dong et al. (2021). Since the base model is the only source of previously seen knowledge, if it is updated, knowledge is lost forever. Typically, existing works decouple the learning of (backbone) feature representations and the classifier by *learning* the model only on the base session data and relying on non-parametric class-mean classifiers for continual few-shot data (Peng et al., 2022; Zhou et al., 2022; Hersche et al., 2022). We differ from these approaches in two aspects: (A) Any non-parametric classifier relies on storing either explicit or class mean feature-prototypes which could raise issues surrounding user identity revelation. (B) Recent works Pernici et al. (2021); Yang et al. (2023) point out the feature-classifier misalignment due to new class prototypes being extracted from a base-class frozen backbone representation. We seek to question if this feature-classifier misalignment dilemma (arising due to training model parameters either only on base or only on continual few-shot classes) can be resolved by input prompt tuning with a parametric classifier, while keeping the feature extractor frozen after base session training.

**Prompt Tuning for Continual Learning.** Conventionally, transfer learning adapts a model to a downstream task by fine-tuning the entire model, referred to as *model tuning*. 'Prefix tuning' Li & Liang (2021) and 'Prompt tuning' Lester et al. (2021) were the first to show that simply training tunable parameters attached to the input of a LLM (and initial network layers) while keeping the backbone frozen is competitive with model tuning. This idea provides a simple and cost-effective way of learning task-specific signal condensed into task-specific 'soft prompts.' A natural alternative to storing privacy violating exemplars and replaying them is to instead train a set of prompts for each sequential continual learning task in the future. The caveat here is that even if one trains a set of prompts for each continual learning task, depending on the continual learning protocol task identity may not be available at inference time (hence the model will not know which task's prompt or classifier to use for evaluating a test sample). In this respect, S-prompts Wang et al. (2022a) and A-la-carte prompt tuning (APT) Bowman et al. (2023) learn an independent set of prompts for each domain/task, using a K-NN based search for domain identity or K-Means classsifier concatenation strategy at inference time. Note that since both S-prompts and APT learn task-specific stand-alone prompts, the prompt feature space is task-specific, and there is no forgetting of old knowledge when learning new tasks. On the other hand, these 'no forgetting' prompts cannot share knowledge across tasks. This leads to another ideology for continual prompting, i.e., treat each prompt unit in the attached prompt as being a part of a larger **shared pool** of prompts. Then the desired number of prompt units can be curated by either hard or soft selection from the pool, conditioned on the input instance itself (Wang et al., 2022c;b; Smith et al., 2023). Given the scarcity of data in FSCIL, we hypothesize that sharing of knowledge will benefit new tasks and draw inspiration from this line of works. We discuss our prompt design differences from these approaches in Section 3.4.
Most recently, Adaptive Prompt Generator (APG) Tang et al. (2023) challenges the intensive ImageNet21K pre-training assumption as it prompts a ViT pre-trained only on the continual benchmark's base class data. They use the knowledge pool prototypes to constraint the prompt and classifier weights for each class by replay- and knowledge distillation-style 'anti-forgetting learning'. Even though our backbone is trained only on the base classes, we propose a simple prompting-only strategy. Finally, we are not the first to attempt continual prompting for temporal data, as PIVOT Villa et al. (2023) trains a temporal encoder on top of a pre-trained CLIP and proposes separate spatial and temporal prompts for image-video continual learning. However, we explore prompting GCNs,

designed to model the spatio-temporal skeletal information in temporal action recognition datasets in a single unified model. This has not been attempted before. Such a prompt strategy is all we need to continually add new action semantics in a few-shot manner.

**Learnable Position Encoding.** Our work can also be thought of as weakly related to the learnable structural and positional encodings (LSPE) for MP-GNNs in (Dwivedi et al., 2022). We perform a simple but effective addition operation to attach our prompt to the input embedding. As our prompts have the same size as the input it can be thought of as a learned prompt encoding, bearing similarity with learnable additive position encoding works (Liu et al., 2020; Li et al., 2021b).

## 3 METHOD

### 3.1 PROBLEM DEFINITION

Few-shot Class Incremental Learning (FSCIL) is the continual learning paradigm where a model sequentially adapts to learn a series of $\mathcal{T}$ training sessions (also called tasks) $\{\mathcal{T}^{(0)}, \mathcal{T}^{(1)}, ..., \mathcal{T}^{(T)}\}$ corresponding to training datasets $\{\mathcal{D}^{(0)}, \mathcal{D}^{(1)}, ..., \mathcal{D}^{(T)}\}$ where $\mathcal{D}^{(t)} = (x_i^t, y_i^t)_{i=1}^{|\mathcal{D}^{(t)}|}$, such that the base session $\mathcal{T}^{(0)}$ has a large label space $\mathcal{Y}^{(0)}$ and contains sufficient training instances $\mid \mathcal{D}^{(0)} \mid$ whereas each subsequent session $\mathcal{T}^{(t)}, t \geq 1$ comprises of only a few training instances ($K$) for each of the $\mid \mathcal{Y}^{(t)} \mid = N$ classes, such that $\mid \mathcal{D}^{(t)} \mid = NK$ and is referred to as a N-way K-shot task. Each session has a disjoint label space $\mathcal{Y}^{(t)} \cap \mathcal{Y}^{(t')} = \emptyset, \forall t \neq t'$. Re-stating our privacy constraints, in any training session $\mathcal{T}^{(t)}$, the model has access to only $\mathcal{D}^{(t)}$ and after training, this data is made inaccessible for use in subsequent sessions. After training on a new session $\mathcal{T}^{(t)}$, the model is evaluated on the test set of all classes seen so far $\cup_{i=0}^{t} \mathcal{Y}^{(i)}$ and the challenge is to alleviate catastrophic forgetting of old classes while preventing overfitting to new classes.

**Preliminaries.** We introduce FSCIL for action recognition such that the input $x \in \mathbb{R}^{T \times J \times C}$ is a temporal sequence of $T$ frames, where each frame is a human skeleton constituted by the set $J = \{\jmath_1, \jmath_2, ..., \jmath_V\}$ of body or hand joints, where $V = \mid J \mid$. Each input joint $\jmath_i$ has a feature dimension $C$ (joint coordinates in 2D or 3D Cartesian system). Such a modality is commonly represented as a graph topology $\mathcal{G}$ and modeled using Message Passing Graph Neural Networks (MP-GNNs) which can either be sparse GCNs (CTR-GCN) or fully connected graph-transformers (DG-STA). Most GNN architectures Dwivedi et al. (2022) can be defined as $f = f_c \circ f_g \circ f_e$, having an input embedding layer $f_e$, a graph feature extractor $f_g$ consisting of a stack of convolutional or attention layers and a fully connected layer at the end for classification $f_c$.

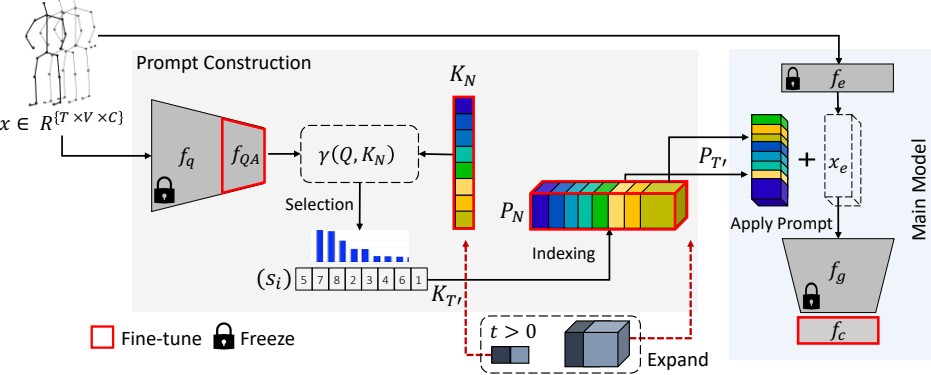

Figure 2: Prompt Offset Tuning (POET): We propose to prompt offset the input feature embedding $x_e$ of the main model by learnable prompt parameters $P_{T'}$ for FSCIL. Construction of $P_{T'}$: An input dependent query $Q$ is matched with learnable keys $K_N$ using sorted cosine similarity. We create a mask of the top $T'$ key indices in the sorted sequence to form $K_{T'}$. This is the final prompt selection which is used to index into the prompt pool $P_N$, forming ordered prompt $P_{T'}$ which we add to $x_e$.

**Prompt Tuning.** The idea of prompting, as it originated from Large Language Models (LLMs) is to add extra information (referred to as a text prompt) to condition the model's input for its generation of the corresponding output label. Instead of applying a discrete, pre-defined 'hard' language prompt token, *prompt tuning* Lester et al. (2021) formalized the concept of applying 'soft prompts' to the input such that a set of learnable parameters is prepended to the input text and is trained by maximum likelihood error backpropagation while keeping the backbone parameters frozen. Transformer based language models and corresponding Vision Transformers (ViT) Dosovitskiy et al. (2020) model an input represented as a sequence of language or image-patch *tokens*. All the continual visual prompting works we refer to in Section 2 prompt a ViT or CLIP backbone.

While the natural choice of prompt tuning for these token-based feature extractors is to prepend (concatenate) the learnable prompt parameters along the token dimension, prompting remains un-explored and undefined (to the best of our knowledge) for non-transformer based backbones such as CNNs and MP-GNNs. Hence, for our work we formally define a set of prompts $P_{T'}$ to be learnable parameter vectors in a continuous space applied to the input feature embedding $x_e$ via a prompt function $f_p$, transforming model input to obtain output logit distribution $y = f(x, P_{T'}) = f_c \circ f_g \circ f_p(f_e(x), P_{T'})$. $x_e$ is produced by the input embedding layer $f_e(x) = x_e, x_e \in \mathbb{R}^{T \times J \times C_e}$ where $C_e$ is the feature dimension. To train on the $| \mathcal{Y}^{(t)} |$ new classes in subsequent sessions $t \geq 1$, the classifier $f_c$ expands to accommodate for the new classes $\theta_c \in \mathbb{R}^{C' \times |\cup_{i=0}^t \mathcal{Y}^{(i)}|}$. Importantly, disparate from most existing continual prompting works, our feature extractor backbone $f_g$ is trained *only on the base class data* $\mathcal{D}^{(0)}$ and will never itself be fine-tuned on action semantics from any of the continual sessions $\mathcal{T}^{(t)}, t > 0$. After the base training session, $f_g$ parameters $\theta_g$ are frozen. We discuss the composition and selection of $P_{T'}$ in Section 3.2, prompt pool expansion in Section 3.3 and prompt attachment and classifier protocols in Section 3.4.

## 3.2 PROMPT DESIGN

$P_{T'} \in \mathcal{R}^{T' \times V \times C_e}$ is an *ordered* set of $T'$ prompts, where each prompt $P_i \in \mathcal{R}^{V \times C_e}$ has a prompt length equal to the dimension of the input skeleton and embedding dimension same as input $x_e$.

As mentioned in Section 2, to encourage knowledge sharing in the data-scarce FSCIL setting, we choose to construct a single prompt pool, sharing encoded knowledge across all tasks such that the sequence of $T'$ prompts is selected from the pool $P_N = \{P_1, P_2, ..., P_N\}, T' \leq N$ to form $P_{T'}$ which is applied to the input feature embedding via $f_p(x_e, P_{T'})$ (refer to Figure 2). For the prompt selection process, we construct a bijective key-value codebook treating prompts in the pool $P_N$ as values and keys $K_N = \{K_1, K_2, ..., K_N\}, K_i \in \mathcal{R}^{C_e}$ are learnable vectors being used to find and fetch relevant (closest) values based on a quantization process. This quantization process involves a query $Q$ and key $K_N$ matching such that the query function $f_q(x)$ is an input dependent deterministic function mapping the input instance to a query with dimensionality same as the key $x \in \mathbb{R}^{T \times J \times C} \to Q \in \mathcal{R}^{C_e}$. Note the selection is input dependent and not task or class dependent, hence we drop subscripts on x. Actual prompt index selection is done as per:

$$\underset{(s_i)_{i=1}^{T'} \subseteq [1,N]}{\arg\max} \quad \gamma(f_q(x), K_N) \tag{1}$$

Where $(s_i)_{i=1}^{T'} = K_{T'}$ is an ordered set of top $T'$ keys sorted by decreasing magnitude of cosine similarity $\gamma(.)$. Indexing (selection) of values from the pool $P_N$ is based on this ordered sequence. We discuss this further in Section 3.3. To move the aligned query-key pairs towards each other, we use a normalized cosine similarity clustering based regularization loss inspired from VQ-VAE Van Den Oord et al. (2017) and similar to L2P (Wang et al., 2022c):

$$\underset{\theta_{QA}, \theta_{K_{T'}}}{\max} \quad \lambda \Sigma_{i=0}^{T'} \gamma(Q, K_N) \tag{2}$$

The query-key based cosine similarity matching is responsible for identification of the relevant 'ordered prompt index sequence', whereas the key-value mapping actually indexes into the pool to curate the final prompt $P_{T'}$ for that input. *So what should be an appropriate choice for $f_q$?* Notice that unlike L2P, we do not have a backbone feature extractor trained on an intensive benchmark dataset encompassing a superset of semantic knowledge the model will potentially see during its lifetime. Assuming access to no other pre-trained backbone, all we have is a query function pre-trained only on the base class data, a copy of the same frozen backbone as the main model $f(x)$, i.e.,

$f_q(x) = f'_g \circ f'_e(x)$ (note classifier layer $f'_c$ will be discarded for $f_q$). Such a $f_q$ is clearly not discriminative enough to select relevant prompts for new classes. Thus, we need a mechanism to update the query on the current task's cross entropy loss so it can stay relevant to new few-shot classes being added. We require the classification error gradients to backpropagate through (i) the selected prompt pool parameters $P_{T'}$, (ii) to the corresponding key parameters that selected the prompts $K_{T'}$ and (iii) tunable parameters in query function $f_q$. Due to the few-shot nature of the problem, fine-tuning the entire $f_q$ will result in completely overfitting to the few shot data and washout of previous class knowledge. Hence, we modify the base data trained query function as $f_q(x) = f_{QA} \circ f'_g \circ f'_e(x)$, where the *query adaptor* $f_{QA}$ is a fully connected layer being updated (starting from base model prompting) to make the query adapt to every new task while freezing all other $f_q$ parameters. We design $f_{QA}$ so that it maps the $f_q$ output dimension $C'$ to the desired input embedding dimension $C_e$ (same as key and prompt feature dimension) in addition to updating the query. Final cross entropy loss for activity and hand gesture recognition can be written as:

$$\min_{\theta_{QA}, \theta_{K_{T'}}, \theta_{P_{T'}}, \theta_{f_c}} \mathcal{L}(f(x, P_{T'}), y) \tag{3}$$

From a practical standpoint, the gradient of cross entropy loss above is not defined for key-query parameters $\theta_{K_{T'}}$ and $\theta_{QA}$ as the argmax operation in Equation 1 generates a non-smooth binary mask of selected key indices $(s_i)$. We approximate the gradient similar to the straight-through estimator reparameterization trick as in Van Den Oord et al. (2017); Bengio et al. (2013), passing the mask during forward pass and copying the gradients from prompt pool to entire sorted cosine similarity during backpropagation.

## 3.3 PROMPT POOL UPDATE IN SUBSEQUENT CONTINUAL TASKS

A crucial observation we make from our initial experiments on is that by construction of the prompt selection process and clustering loss in Equation 2, when the randomly initialized keys and prompts are first trained in the base step $\mathcal{T}^{(0)}$, they select a random set of prompts and start optimizing them (only the indexed prompt parameters, i.e. $P_{T'}, K_{T'}$ appear in the backpropagation computation graph). We find that this leads to selection of the same set of prompts in all subsequent steps, as the clustering brings the queries $Q$ close to the selected keys $K_{T'}$. But, our ordered prompt selection is discriminative for most examples such that certain prompts tend to appear before others in the sorted sequence for different task semantics.

Building on this observation, we simplify prompt selection in $\mathcal{T}^{(0)}$ such that $N = T'$ for the base session prompt pool. Hence, in the base session, all prompts in the pool are selected but in *sorted order of their relevance for each input instance*. Interestingly, this order helps achieve high plasticity on the new tasks, as the scarcely supervised new classes benefit heavily from the pre-existent knowledge being accumulated from previous tasks. While this is desirable for performance on the new task, it leads to interference with the old tasks as the same prompt parameters are still being optimized on each subsequent task's few-shot training samples. To address this challenge, we propose to **expand** the pool by adding a set of $R$ new prompts (Algorithm 1). We demonstrate that this leads to a significant improvement in new task performance, while not affecting old task performance. We select $T'$ freely from all prompts $P_{N+R}$, without task identity at inference time.

---

**Algorithm 1** Prompt Pool Expansion at Training Time, for $t \geq 1$

---

**Require:** $P_N, K_N$ from previous session $t - 1$
  Expand pool and key by $R$ new prompts as: $P_N \rightarrow P_{N+R}$; $K_N \rightarrow P_{N+R}$
  Where $P_{N+R} = \{P_N; P_R\}$ (concatenate new prompts at the end of sequence, explicitly use them)
  Initialize parameters of new $P_i \leftarrow Mean(P_N)$ and new $K_i \leftarrow Mean(K_N)$
  **Construct** $P_{T'}$ as
  1. Select $T' - R$ key indices $K_{T'-R}$ from previous pool $P_N$ as per Eq 1 to form $P_{T'-R}$
  2. Concatenate new prompts to this to make $P_{T'} = \{P_{T'-R}; P_R\}$
  Freeze previous task prompts in the pool $P_N$
  **Train** only new prompts $P_R$ and all keys $K_{N+R}$ (to learn inter-task selection)

---

## 3.4 DESIGN DIFFERENCES FROM EXISTING WORKS

Our key-value codebook style prompt selection mechanism is directly inspired from vector quantization works VQ-VAE Van Den Oord et al. (2017), and existing shared prompt-pool based continual

learning works, (i) L2P Wang et al. (2022c) and (ii) discrete key-value bottleneck Träuble et al. (2023). However, we observe three key differences as compared to these approaches:

- L2P focuses on decoupling the prompt selection process (query-key matching) from the actual prompt pool parameters being selected $P_{T'}$. More concretely, they update key parameters only on the clustering loss in Equation 2, while the classification error gradients backpropagate only till the prompt pool, as the argmax operation for selecting $K_{T'}$ creates non-smooth neurons (note they do not have a $QA$). As opposed to them, our coupled design of the prompt key, value, and query adaptor helps update all $P_{T'}, K_{T'}, QA$ parameters so as to make the prompt selection process better adapted to previously unseen classes. As an additional benefit, it also helps control prompt expansion by making the key better suited to globally select from the expanded pool Section 3.3. We demonstrate in our experimental section that both decoupled prompting (L2P) and soft prompting (CODA-P) does not work well for our setting.
- $P_{T'}$ is not just a set of selected top $T'$ prompt parameters being used to transform the input, but is an ordered temporal sequence of continuous learnable parameters, selected by indexing into a shared prompt pool based on the sorted query-key cosine similarity for each input instance. This can be thought of as a self-regularization as different input semantics will select discriminative prompt ordering to be applied for transforming it. The importance of the ordered sequence selection is further validated by the difference in attaching new expanded prompts $R$ before versus after in the sequence when training with expanded prompt pool and keys in $t \geq 1$. We hypothesize that the ordering is such that the frames occurring in the beginning of the sequence are learning more task-agnostic features while appending new prompts towards the end helps learn task-specific features in those frames.
- We empirically analyse the choice of prompt transformation function $f_p$ for our temporal skeleton modality. As compared to all existing prompt tuning works which concatenate a set of prompts along the token dimension, we find that a naive addition operation works better as compared to concatenation along temporal dimension or rolled out spatio-temporal dimension (we try both). We also discover that selecting the same number of prompts as the input temporal dimension $T' = T$ yields best results, providing a learnable transformation for each joint in the spatio-temporal graph embedding. The side benefit of this observation is our proposed prompt strategy is a frustratingly simple plug-and-play prompt such that a learnable parameter having size same as the input feature embedding size needs to be added to the model input feature, making it invariant to datasets, backbones and input sizes, as: $f_p(x_e, P_{T'}) = x_e + P_{T'}, T' = T$

We show the significance of each of these design components via ablations in Section 4.3.

## 4 EXPERIMENTS

### 4.1 DATASET SPLITS AND EXPERIMENTAL PROTOCOL

We set up two new few-shot continual action recognition benchmarks, one on 3D skeleton-based Activity Recognition using the NTU RGB+D dataset Shahroudy et al. (2016) and another on 3D skeleton-based Hand Gesture Recognition using the SHREC-2017 dataset Smedt et al. (2017). It is important to note that while we propose a prompt-based solution strategy, our dataset split protocols follow the pre-prompting conventional continual learning experimental dataset splits, wherein base task is trained on a subset of the benchmark dataset itself (and not any large pre-training dataset as done in L2P and CODA-P). The model being prompted is seeing every semantic class only once, and every new session has new data. For NTU RGB+D dataset, we divide the 60 action classes into 40 base and 20 few-shot continual classes. We propose a 4-task 5-way 5-shot (N-way K-shot) protocol, such that 5 new classes are being learning over 4 sequential tasks, each having only 5-training instances per class. For SHREC-2017, we divide the 14 hand gesture classes into 8 base classes and 6 continual classes are learnt sequentially in a 3-task 2-way 5-shot protocol. We cover all implementation details in Appendix A.1.

### 4.2 EVALUATION METRICS

Following Peng et al. (2022), we report Harmonic Mean accuracy $A_{HM}$ in addition to the Average accuracy after learning every new task. Note that the average accuracy tends to be biased towards base session $\mathcal{T}^{(0)}$ performance due to more number of base classes. A higher $A_{HM}$ implies better

stability-plasticity trade-off between new task performance and old tasks' retention. A lower average and higher $A_{HM}$ indicates better plasticity. Unlike prior continual learning works, we dive into stability-plasticity trade-offs by transparently analysing 'Old' and 'New' accuracy. The 'Old' is an average of all previously seen classes.

### 4.3 INSIGHTS INTO OUR PROMPT-OFFSET TUNING

**Prompt pool design ablations.** We compare the contribution of individual prompt design elements in POET using ablation Table 1. 'FE' denotes the standard continual learning baseline 'Feature Extraction' where the model backbone is frozen, while only expanding and updating the classifier on new classes. As our prompt tuning mechanism is designed on top of FE, this shows that if we completely remove prompts from our method, there will be a $19.5\%$ drop in average accuracy and $15.7\%$ drop in Harmonic Mean. Notice our proposed prompting improves both old and new task performance. Next, the 'Decoupled' prompting experiment indicates the importance of our coupled prompt selection mechanism (using reparameterization trick and query function update). This is a direct comparison of our additive *prompt attachment* with the decoupled key-value *prompt selection* in Learning to Prompt Wang et al. (2022c). Only not updating query adaptor $QA$ reduces plasticity of Task 4 by $4.6\%$. Not expanding the prompt pool by $R$ new prompts brings down 60 class $A_{HM}$ by $2.2\%$. Not freezing previous prompt pool parameters during pool expansion reduces previous task average by $2.8\%$ by the final task. Interestingly, as we mentioned in Section 3.4 point 2, attaching new prompts at the beginning of the ordered prompt sequence reduces new task performance by $3.3\%$ as compared to attaching it at the end of the cosine-similarity selected $T' - R$ prompts. This validates that our method is indeed learning order discriminability. Note, for all these ablation experiments, we curate and *add* 64 prompts from the pool, keeping prompt attachment constant and only varying other components.

Table 1: Ablation analysis of our prompt-offset tuning, on NTU RGB+D Activity Recognition

| | T0 | T0 → T1 | | | {T0, T1} → T2 | | | {T0, T1, T2} → T3 | | | {T0, T1, T2, T3} → T4 | | | |
|---|---|---|---|---|---|---|---|---|---|---|---|---|---|---|
| Method | Base | Old | New | Avg | Old | New | Avg | Old | New | Avg | Old | New | **Avg** | $A_{HM}$ |
| POET (Ours) | 87.9 | 84.7 | 66.0 | 82.6 | 80.1 | 45.8 | 77.0 | 69.9 | 59.2 | 68.8 | 59.3 | 57.4 | 59.4 | 58.4 |
| FE (No Prompting) | 88.4 | 76.5 | 59.1 | 74.5 | 67.9 | 51.4 | 66.3 | 50.4 | 40.8 | 49.5 | 39.2 | 46.8 | 39.9 | 42.7 |
| Decoupled L2P-style | 88.0 | 85.3 | 61.6 | 82.8 | 78.0 | 50.6 | 75.3 | 67.1 | 55.1 | 65.8 | 56.5 | 51.3 | 56.1 | 53.8 |
| w/o clustering loss | 85.5 | 86.6 | 41.5 | 81.6 | 78.9 | 36.8 | 74.3 | 69.2 | 28.9 | 64.5 | 62.0 | 18.2 | 57.0 | 28.1 |
| w/o QA update | 87.9 | 84.7 | 66.3 | 82.8 | 80.9 | 45.5 | 77.4 | 70.1 | 59.5 | 69.1 | 59.4 | 52.8 | 58.7 | 55.9 |
| w/o Pool Expansion | 87.9 | 84.7 | 66.6 | 82.6 | 80.5 | 45.3 | 77.0 | 69.9 | 58.3 | 68.9 | 60.1 | 52.8 | 59.5 | 56.2 |
| w/o Freezing Previous Prompts | 87.9 | 84.8 | 67.1 | 82.9 | 80.8 | 44.9 | 77.3 | 70.1 | 59.5 | 69.1 | 59.5 | 54.6 | 58.6 | 56.9 |
| Attach new prompts before | 87.9 | 84.8 | 66.6 | 82.7 | 80.7 | 45.8 | 77.2 | 70.2 | 59.2 | 68.9 | 59.3 | 54.1 | 58.8 | 56.6 |

**Prompt attachment.** As mentioned in Section 3.1 the dimensionality of a temporal prompt and the prompt function $f_p$ are not well defined for non-transformer based architectural backbones. We empirically explore potential choices for $f_p$ in Table 2. Drawing a parallel with sequence based transformers which concatenate prompts along the token length, we concatenate prompts along the temporal dimension of the skeleton input feature embedding $x_e$. Since addition of prompt $P_{T'}$ to $x_e$ yields the best result, we call our prompt tuning solution as 'Prompt Offset Tuning' (POET).

Table 2: Prompt Function $f_p$ Exploration, empirical analysis of prompt attachment operations. Note, we attach 64 prompts selected by our coupled pool selection method in all these experiments.

| | T0 | T0 → T1 | | | {T0, T1} → T2 | | | {T0, T1, T2} → T3 | | | {T0, T1, T2, T3} → T4 | | | |
|---|---|---|---|---|---|---|---|---|---|---|---|---|---|---|
| Method | Base | Old | New | Avg | Old | New | Avg | Old | New | Avg | Old | New | **Avg** | $A_{HM}$ |
| Addition (Ours) | 87.9 | 84.7 | 66.0 | 82.6 | 80.1 | 45.8 | 77.0 | 69.9 | 59.2 | 68.8 | 59.3 | 57.4 | 59.4 | 58.4 |
| Concatenate, temporal dim | 88.6 | 69.7 | 75.6 | 70.3 | 62.5 | 60.2 | 62.4 | 48.7 | 60.0 | 49.8 | 33.6 | 50.5 | 35.1 | 40.3 |
| Concatenate, feature dim | 87.7 | 85.4 | 58.4 | 82.4 | 79.2 | 41.8 | 75.5 | 68.2 | 54.2 | 66.9 | 57.09 | 41.5 | 56.0 | 48.1 |

### 4.4 COMPARISON WITH STATE-OF-THE-ART

In Fine-tuning (FT) all model parameters are trained on new task data, FSCIL is quite challenging for this modality as old task performance reduces to zero from T1 itself. Feature Extraction freezes the backbone while only updating the classifier (after expanding the classifier for each new task). In FE frozen, we freeze parts of the classifier parameters that belong to previous classes to

prevent forgetting from the classifier. FE replay indicates feature extraction with storage and replay of the incremental few-shots. Note that our setting prohibits base data (proprietary company data) replay. FE and FT experiments tell us that the 8-class SHREC, DGSTA backbone is highly plastic and the 40-classs NTU60, CTRGCN backbone is more stable. Next we implement knowledge distillation based LwF, EWC, replay and LUCIR all of which require fine-tuning of the backbone with additional regularization losses. Their failure indicates that any kind of backbone fine-tuning is not feasible for FSCIL. We compare with three continual prompting approaches, L2P, CODA-P and APT. We also compare with one of the newer FSCIL baselines (ALICE), originally developed for image benchmarks on SHREC Table 4. Note the high retention of base task performance, but at the same time poor plasticity and adaptation to new task classes - this is the issue of feature-classifier misalignment that we hope to alleviate through prompt tuning. Note that the joint (offline, non-sequential) baseline in Table 3 is outperformed by 3.7% by the corresponding prompted joint model. This, when compared with the Base (T0) performance with and without prompting shows that perhaps prompting is particularly useful for learning few-shot data, as the few-shot data leverages the class-agnostic knowledge from classes with full supervision using prompts.

Table 3: Performance of few-shot continual learning on our NTU RGB+D Shahroudy et al. (2016) Activity Recognition benchmark, based on CTR-GCN backbone. Numbers indicate Test Accuracy (%, ↑). After training on each incremental task, we report (A) performance of all old classes averaged ('Old'), (B) only new classes ('New'), and (C) Average of all classes seen ('Avg') to understand stability-plasticity trade-offs and $A_{HM}$ in the last task.

| Method | T0 Base | T0 → T1 Old | New | Avg | {T0, T1} → T2 Old | New | Avg | {T0, T1, T2} → T3 Old | New | Avg | {T0, T1, T2, T3} → T4 Old | New | **Avg** | $A_{HM}$ |
|---|---|---|---|---|---|---|---|---|---|---|---|---|---|---|
| Joint | 88.4 | | | 79.0 | | | 71.0 | | | 66.8 | | | 63.5 | |
| Joint Prompt | | | | | | | | | | | | | **67.2** | |
| FE | 88.4 | 76.5 | 59.1 | 74.5 | 67.9 | 51.4 | 66.3 | 50.4 | 40.8 | 49.5 | 39.2 | 46.8 | 39.9 | 42.7 |
| FE, frozen | 88.4 | 81.7 | 49.2 | 78.1 | 64.7 | 40.0 | 62.3 | 43.2 | 27.6 | 41.7 | 29.5 | 25.5 | 29.2 | 27.4 |
| FT | 88.4 | 0.4 | 58.2 | 6.8 | 0.0 | 59.2 | 6.0 | 0.0 | 41.6 | 3.8 | 0.0 | 25.1 | 2.1 | 0.0 |
| FE, replay** | 88.4 | 73.5 | 70.1 | 73.1 | 59.2 | 71.0 | 60.4 | 61.8 | 59.4 | 61.6 | 60.3 | 28.8 | 57.7 | 39.0 |
| LWF (Li & Hoiem, 2017) | 88.4 | 0.5 | 74.0 | 8.7 | 0.0 | 31.5 | 3.2 | 0.0 | 31.4 | 2.9 | 0.0 | 23.1 | 1.9 | 0.0 |
| EWC (Kirkpatrick et al., 2017) | 88.4 | 0.0 | 57.4 | 6.4 | 0.0 | 62.2 | 6.2 | 0.0 | 33.9 | 3.1 | 0.0 | 28.9 | 2.4 | 0.0 |
| Experience Replay** | 88.4 | 0.4 | 59.8 | 7.0 | 6.1 | 5.2 | 6.0 | 6.8 | 42.2 | 10.0 | 10.1 | 32.2 | 12.0 | 15.4 |
| LUCIR (Hou et al., 2019) | 87.9 | 0.0 | 19.9 | 2.2 | 0.0 | 20.1 | 2.0 | 0.0 | 29.2 | 2.7 | 1.3 | 2.5 | 1.4 | 0.0 |
| CODA-P (Smith et al., 2023)* | 87.4 | 86.1 | 4.0 | 76.3 | 76.0 | 0.3 | 68.4 | 65.9 | 0.2 | 59.7 | 60.9 | 0.7 | 55.7 | 1.3 |
| CODA-P (Smith et al., 2023) | 87.6 | 4.9 | 7.4 | 4.5 | 31.7 | 0.0 | 28.8 | 0.5 | 1.6 | 0.6 | 35.4 | 0.7 | 31.1 | 1.3 |
| L2P (Wang et al., 2022c)* | 88.6 | 88.4 | 0.0 | 78.5 | 78.5 | 0.0 | 70.7 | 70.1 | 0.0 | 63.7 | 61.2 | 0.0 | 56.2 | 0.0 |
| L2P (Wang et al., 2022c) | 84.7 | 3.0 | 0.4 | 2.7 | 3.1 | 0.0 | 2.8 | 2.8 | 0.0 | 2.6 | 2.4 | 0.4 | 2.2 | 0.6 |
| APT (Bowman et al., 2023)^ | 86.6 | | 29.2 | | | 30.6 | | | 35.9 | | | 35.0 | | |
| POET (Ours) | 87.9 | 84.7 | 66.0 | 82.6 | 80.1 | 45.8 | 77.0 | 69.9 | 59.2 | 68.8 | 59.3 | **57.4** | **59.4** | **58.4** |

Table 4: Performance on SHREC-2017 Smedt et al. (2017) Gesture Recognition Dataset benchmark. Reporting mean and STD across 5 runs.

| Method | T0 Base | T0 → T1 Old | New | Avg | {T0, T1} → T2 Old | New | Avg | {T0, T1, T2} → T3 Old | New | **Avg** | $A_{HM}$ |
|---|---|---|---|---|---|---|---|---|---|---|---|
| Joint | 88.75 | 91.2 ± 0.4 | 37.5 ± 3.0 | 79.4 ± 0.7 | 91.4 ± 1.1 | 52.2 ± 4.2 | 77.3 ± 2.1 | 90.8 ± 0.2 | 47.6 ± 2.8 | 70.9 ± 1.2 | 62.4 ± 0.4 |
| FT | 88.75 | 0.0 | 91.6 ± 3.7 | 20.3 ± 0.8 | 0.0 | 69.5 ± 11.9 | 12.4 ± 2.1 | 0.0 | 85.8 ± 9.4 | 13.4 ± 1.5 | 0.0 |
| FE | 88.75 | 56.2 ± 3.4 | 85.7 ± 6.5 | 62.7 ± 2.4 | 37.2 ± 6.7 | 64.3 ± 11.9 | 41.9 ± 6.9 | 17.5 ± 5.1 | 77.3 ± 8.8 | 26.8 ± 3.4 | 28.5 ± 6.4 |
| FE, frozen | 88.75 | 69.6 ± 4.1 | 77.4 ± 9.2 | 71.3 ± 1.9 | 61.9 ± 2.1 | 59.1 ± 11.7 | 61.4 ± 2.7 | 44.7 ± 3.2 | 54.5 ± 6.7 | 46.2 ± 2.7 | 49.1 ± 4.3 |
| LWF (Li & Hoiem, 2017) | 88.75 | 0.0 | 91.2 ± 6.2 | 20.2 ± 1.4 | 0.0 | 70.3 ± 5.8 | 12.5 ± 1.0 | 0.0 | 88.4 ± 13.7 | 13.8 ± 2.1 | 0.0 |
| L2P (Wang et al., 2022c) | 88.77 | 17.3 ± 5.2 | 30.9 ± 25.9 | 20.3 ± 5.9 | 12.3 ± 5.9 | 2.1 ± 3.8 | 10.5 ± 4.8 | 8.2 ± 4.0 | 6.9 ± 8.5 | 7.9 ± 3.9 | 7.5 ± 5.5 |
| CODAP (Smith et al., 2023) | 87.67 | 19.9 ± 5.6 | 0.5 ± 0.7 | 15.6 ± 4.5 | 13.6 ± 4.0 | 2.2 ± 1.9 | 11.6 ± 1.9 | 7.9 ± 1.8 | 14.1 ± 21.4 | 8.8 ± 2.4 | 10.1 ± 3.2 |
| ALICE (Peng et al., 2022) | 92.07 | 86.0 ± 3.5 | 24.5 ± 14.9 | 72.4 ± 5.7 | 72.1 ± 5.7 | 22.5 ± 16.9 | 63.3 ± 7.6 | 62.5 ± 6.8 | 11.9 ± 9.9 | **54.6 ± 6.9** | 20.0 ± 8.1 |
| POET (Ours) | 91.85 | 71.1 ± 4.7 | 80.8 ± 6.7 | 73.2 ± 3.7 | 63.1 ± 2.2 | 56.7 ± 9.4 | 61.9 ± 1.8 | 45.9 ± 2.6 | 72.4 ± 7.1 | 50.0 ± 1.6 | **56.2 ± 1.6** |

## 5 CONCLUSION

We show that FSCIL for activity recognition can be solved using just a frozen backbone trained only on base-class data and stability-plasticity trade-offs can be attained by relying solely on a prompt-tuning strategy with classifier expansion, without use of any other conventional continual learning methods such as knowledge distillation, prior-based regularization, rehearsal or parameter isolation. Hence, we demonstrate the merit of prompting for a real-world continual learning problem setting for a non-image input modality and hope that it becomes the preferred choice for continual learning solutions even in applications where a well pre-trained backbone is unavailable.

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
