# A  APPENDIX

## A.1  IMPLEMENTATION DETAILS

As mentioned in Section 3.1, MP-GNNs can either be sparse or fully connected. We demonstrate our method on both types of architectures. For 3D skeleton-based activity recognition, we prompt the sparse Graph Convolutional Network CTR-GCN Chen et al. (2021). NTU RGB+D has 40,320 training and 16,560 testing samples. We use the cross-subject evaluation protocol, with only-joint input modality for ease of prompting. Each input skeleton sequence has 64 temporal frames, each consisting of 25 body keypoints, such that $x \in \mathcal{R}^{64,25,3}$. We start from a prompt pool of size $N = 64$ and expand by 6 prompts. To develop the prompt expansion protocol, we created a separate 30% validation set of only $t \geq 1$ classes. For 3D skeleton-based hand gesture recognition, we prompt a fully-connected graph-transformer architecture, DG-STA Chen et al. (2019). This is a small-scale dataset with 1980 training and 840 testing samples, each containing temporal sequences of 22 hand joints. We use 8 temporal sequences, such that input $x \in \mathcal{R}^{8,22,3}$, expanding an initial pool of 8 prompts by 2 in each continual session.

We plan to release code, models and training few-shots for both NTU RGB+D and SHREC 2017 if/when work is accepted. In accordance with parallel work in image-based FSCIL benchmarks, we conduct experiments on a fixed set of 5-shots per each continual class selected for all experiments on NTU RGB+D. On SHREC, each base model class has only 145 training samples with 8 classes being learnt. The FSCIL performance depends on the selection of few-shots. Hence, we conduct experiments on 5 sets of 5-shots and report mean and standard deviation across all 5 few-shot seed runs. The clustering loss coefficient $\lambda = 0.1$ for all experiments. Learning rate of $\theta_{QA} = 0.01$. Base learning rate for CTRGCN is 0.1 (trained for 50 epochs). Each incremental task is trained for 5 epochs with an initial learning rate of 0.1, we select this because reducing the learning rate renders the model completely unable to learn new knowledge. For DGSTA, base model is trained for 300 epochs starting LR=0.001, while incremental tasks are trained for 30 epochs starting LR=0.01 (to render plasticity for learning new knowledge).

For adapting CODA-P, L2P for CTRGCN, we concatenate along temporal dimension. For DGSTA, we concatenate along rolled out spatio-temporal dimension $(8 * 22)$ as its a fully connected graph transformer and every keypoint can be considered to be a token. In the latter, we use a FC layer to rescale the dimension because we require that the size of the input not be changed otherwise the internal attention layers would have to be rethought.

Existing CIL benchmarks for action recognition and why we don't use their dataset splits: The experimental protocol in Li et al. (2021a) involves a single continual session, learning from 50 base classes to 10 new classes at once which we consider to be a less realistic continual setting to study real-world FSCIL on. The latest data-free CIL hand gesture work Aich et al. (2023) can be too challenging a benchmark to study FSCIL on as there are typically only 5-shots for training each continual class. To the best of our knowledge, FSCIL for skeleton based action recognition has not been attempted previously.

## A.2  FORGETTING FROM THE CLASSIFIER

Unlike most existing FSCIL works which learn non-parametric class-mean classifiers and do not expand the classifier with any new weights, we need to expand and train the classifier in order to obtain the error gradients for updating the prompts. We observe that a major source of forgetting is from the classifier as the logits tend to become heavily biased towards the few-shot new classes. For SHREC - DGSTA, we find that a freezing of previous class weights in classifier works well. We do this by making the gradients of those parameters zero. If entire classifier is fine-tuned, forgetting is exacerbated because the backbone is trained on a smaller scale dataset as compared to NTU RGB+D. For activity, finding a right classifier was not as simple as zeroing out the gradients of part of the classifier parameters. We replace $f_c$ with a cosine normalization classifier $\theta_c^T$:

$$p(x) = \frac{exp(\eta < \theta_{c_i}^T f_g(x) >)}{\Sigma_j exp(\eta < \theta_{c_j}^T f_g(x) >)} \quad (4)$$

The results are in Table **??**, the cosine normalization prevents the complete forgetting of the new-old classes that were trained previously using only few shots. Also, for incremental sessions in

CTRGCN, we initialize the expanded new-class parameters in the classifier as mean of previous class parameters. Such an initialization trick is a must in any FSCIL work.

Table 5: Classifier Analysis for NTU60

| | T0 | T0 → T1 | | {T0, T1} → T2 | | | {T0, T1, T2} → T3 | | | | {T0, T1, T2, T3} → T4 | | | | |
|---|---|---|---|---|---|---|---|---|---|---|---|---|---|---|---|
| Activity | T0 | T0 | T1 | T0 | T1 | T2 | T0 | T1 | T2 | T3 | T0 | T1 | T2 | T3 | T4 | Avg |
| Regular, freeze | 89.2 | 73.5 | 72.5 | 71.3 | 2.1 | 63.8 | 65.3 | 0.0 | 0.0 | 52.9 | 58.7 | 0.0 | 0.0 | 0.0 | 48.9 | 43.1 |
| Regular, tune | 89.2 | 80.9 | 64.9 | 67.4 | 22.1 | 34.4 | 57.2 | 6.32 | 24.3 | 26.9 | 45.0 | 4.6 | 20.9 | 13.3 | 21.4 | 35.2 |
| Cosine classifier | 87.9 | 84.9 | 65.6 | 83.2 | 56.0 | 45.8 | 78.2 | 36.3 | 34.5 | 59.9 | 71.3 | 18.4 | 19.8 | 46.2 | 57.4 | 59.2 |

## A.3 PROMPT ATTACHMENT LAYER

By default, we apply the prompts to the feature embedding after L1 of CTRGCN. In Table 5, we conduct analysis attaching the prompts to deeper layers instead of L1 and find a drop, indicating this is indeed input prompt tuning, where learnable parameters need to be attached to the input embedding, not intermediary or later layers.

Table 6: Prompt Attachment Layer

| | T0 | T0 → T1 | | | {T0, T1} → T2 | | | {T0, T1, T2} → T3 | | | {T0, T1, T2, T3} → T4 | | | |
|---|---|---|---|---|---|---|---|---|---|---|---|---|---|---|
| Method | Base | Old | New | Avg | Old | New | Avg | Old | New | Avg | Old | New | **Avg** | $A_{HM}$ |
| layer 1 (Ours) | 87.9 | 84.7 | 66.0 | 82.6 | 80.1 | 45.8 | 77.0 | 69.9 | 59.2 | 68.8 | **59.3** | **57.4** | **59.4** | **58.4** |
| layer 2 | 88.3 | 83.3 | 67.7 | 81.7 | 77.8 | 56.0 | 75.7 | 70.0 | 57.7 | 68.8 | 58.8 | 54.1 | 58.5 | 56.3 |
| layer 3 | 88.4 | 85.0 | 60.0 | 82.3 | 79.3 | 52.2 | 76.6 | 70.9 | 54.1 | 69.3 | 60.1 | 48.9 | 59.1 | 53.9 |

## A.4 FUTURE WORK

We propose a novel **FSCIL prompting** strategy that continually extends its knowledge over time without access to a large well-trained 'generalist' backbone. However, originally prompt tuning was designed as a means of separating the 'generalist parameters' of a large language model from the 'task-specific prompts' learnt for each transfer learning task Lester et al. (2021). Hence, it will be interesting to show how our method and its various design choices adapt if (A) a 'generalist base backbone' trained on a large corpus of action recognition data becomes available in the future. This backbone should ideally encompass a broad range of action semantics that can potentially be encountered in the model's lifetime as part of continual tasks, such that the actual continual learning benchmark data has mere distribution shifts but is not completely *new* knowledge as done in most existing continual visual prompting works (Wang et al., 2022a; Smith et al., 2023; Wang et al., 2022c; Villa et al., 2023). (B) Also, prompt tuning has been shown to become highly competitive with scale Lester et al. (2021) and prompting with alignment of multiple modalities is effective in alleviating the under specification of few-shot learning Lin et al., 2023. Hence, it will be interesting to develop such large multi-modal backbones combining various input modalities such as 3D skeleton, 2D skeleton, video-action data and parametric body model datasets, for FSCIL prompting of action-related tasks without having to train separate backbones for each modality.