# OpenReview forum: "POET: Prompt Offset Tuning for Continual Few-Shot Action Recognition"
_ICLR.cc/2024/Conference — ICLR 2024 Conference Withdrawn Submission_

### Official Review · Reviewer_ioCA · 2023-10-27

**Soundness:** 2 fair
**Presentation:** 3 good
**Contribution:** 2 fair
**Rating:** 3
**Confidence:** 4

**Summary:**

This paper formalizes a problem setting named privacy aware few-shot class incremental learning for activity and gestures. A Prompt-offset Tuning (POET) method is proposed to solve this problem, which includes a temporal-ordered learnable prompt selection module. Experimental results on action recognition and gesture recognition indicate the effectiveness of the proposed method.

**Strengths:**

- A new problem setting named privacy aware few-shot class incremental learning for activity and gestures is proposed.
- A prompt-based method is introduced to solve the proposed setting.

**Weaknesses:**

- The motivation for the proposed POET is not clear, which makes this paper like an incremental work. Authors should show some properties of POET that fit well into the FSCIL task. Rather than simply applying existing prompt learning techniques to the proposed FSCIL task.
- The proposed prompt-based method seems like a general approach. Can POET be used to solve other continual learning tasks like L2P?
- It is not clear why pretrained weights like ImageNet21K pretraining cannot be used. This greatly limits the application value of the research.
- Qualitative experimental analysis should be provided to facilitate understanding of the properties of the proposed method.

**Questions:**

- How does the proposed method compare to existing methods in scenarios where pre-training is available, such as ImageNet pretraining?

---

### Official Review · Reviewer_SdtP · 2023-10-30

**Soundness:** 2 fair
**Presentation:** 2 fair
**Contribution:** 2 fair
**Rating:** 3
**Confidence:** 3

**Summary:**

This paper proposes a prompt-offset tuning.

They form a new problem setting called privacy aware few-shot class incremental learning for activity and gestures.

The approach uses the temporal sequencing in the data stream of actions and gestures to propose a unique temporal-ordered learnable prompt selection and prompt attachment.

They evaluate the proposed method on two benchmarks, NTU RGB+D and SHREC-2017 for action recognition and hand gesture recognition respectively.

**Strengths:**

+ Overall the paper is technical sound.

+ There are some nice and interesting comparisons presented in the experimental section.

**Weaknesses:**

Major:

-  Formulas: In the approach section, the author gives a detailed description of every mathematical detail, which is worth encouraging, but neglects the main idea and intention behind the module design. To be honest, I am lost in a large number of mathematical symbols. I don't know why these modules are designed in this way and what the structure does.

-  It is suggested to have a notation section detailing the Maths symbols used in the paper, for, e.g., vectors, matrices, tensors, etc.

- Very limited experiments / evaluations (i) the datasets are limited to 2 smaller datasets (ii) the scenarios are only limited to action recognition and gesture recognition, one dataset each. This is a noteworthy shortcoming.

- The ablation studies in section 4.3 are not explained clearly. The analysis and discussions are very limited and not clearly presented to readers. It is suggested to add detailed descriptions for e.g., variants, experimental settings, and provide more insights to show the patterns and explain clearly why.

-  Figures: The research paper should use sufficient figures to show the details of the model, and also visualizations of experimental results in different formats. In this paper, the author only draws two figure, and all the other details are contained in plain texts and tables. Why there are no any visualizations for experimental sections.

- Description: Research papers should primarily focus on effectively describing their own innovations. An exceptional paper should strive to clarify ideas for readers, avoiding vague or unclear presentations. The approach section appears to be poorly written and lacks fluency in its flow. The expression in English should be enhanced to make the paper more engaging and less dry.

-  Table 4 what does 0.0 performance mean?

Minor:

- Abstract section, AR and VR should be explained properly and clearly rather than simply using abbreviations.

- The left quotation marks are all not properly handled, e.g., page 2, 3, especially page 8 section 4.3.

**Questions:**

Please refer to weakness section.

---

### Official Review · Reviewer_H7jd · 2023-10-31

**Soundness:** 3 good
**Presentation:** 2 fair
**Contribution:** 2 fair
**Rating:** 5
**Confidence:** 4

**Summary:**

This paper introduces prompt-offset tuning (POET) for continual few-shot skeleton-based action and hand gesture recognition. It aims at learning the novel classes with few training samples incrementally after training the model on base classes to protect the privacy. Specifically, it firstly trains the model on base classes and freeze it afterwards. Then a prompt module is built to handle the novel classes inspired by LLM prompt tuning. In this way, the whole network does not need to be fine-tuned since the prompt module can efficiently tackle the new classes. In addition, two benchmarks on NTU-RGB+D 60 and SHREC-2017 are built for corresponding evaluation. The experiment results show that the proposed method has good performance.

**Strengths:**

1) This paper introduces a new task, namely privacy aware few-shot class incremental learning for activity and gestures.
2) The idea of adapting LLM prompt tuning for few-shot action recognition is novel and makes sense.
3) The components of prompt module are designed with clear goals
4) The experiment results and ablation studies demonstrate the effectiveness of the method.

**Weaknesses:**

1) In the introduction, the paper talks about the ViT pretrained on ImageNet. It claims one advantage of POET is that it works well without pretrained backbone. However, 3D skeletons are pretty precise and representative data compared to images. Most graph convolution based action recognition methods do not need a pretrained backbone to extract features. To demonstrate the claim, a comparison of with and without pretrained backbone on video-based dataset is more convincing. (If I didn’t misunderstand, the “backbone” in the following sections refers to the model trained on the base classes, which is different from the concept in the introduction)
2) The proposed benchmarks miss many details. The NTU-RGB+D 60 dataset is divided into 40 base and 20 few-shot continual classes. Usually, we follow the cross-subject or cross-view settings so that the model generalization can be evaluated. Same for the SHREC-2017 dataset. Classes that are used for training and testing should be specified so that future work can follow the same settings to make comparison.
3) The majority of the model is frozen after training on the base classes, so there should be a quantitive comparison to demonstrate the computation efficiency improvement and the reduction of trainable parameters when learning the novel classes.
4) If the f_q is fine-tuned on both base and few-shot classes, there will not be overfitting for the few shot data.

**Questions:**

1) The “skeleton-based action recognition” may be more accurate for the title.
2) I am confused about the privacy aware part of the method. The model is trained incrementally with all the novel classes. Even each novel dataset is discarded after training, all the novel data are still used for training and the model contains the information of all trained classes. How is the privacy protected?
3) For the other methods, how are they implemented? E.g. what are the architectures used and additional regularization losses mentioned in page 9?
4) In Table 3, why there are two “CODA-P”s and two “L2P”s? What are the differences?
5) The paper claims it is better than knowledge distillation, prior-based regularization, rehearsal or parameter isolation. Is there are comparison?
6) There are ”??” in page 13, please fix it

**Details Of Ethics Concerns:**

No ethics concerns

---

### Official Review · Reviewer_c8Px · 2023-10-31

**Soundness:** 2 fair
**Presentation:** 3 good
**Contribution:** 2 fair
**Rating:** 3
**Confidence:** 5

**Summary:**

In this paper, the author introduced a few-shot class-incremental learning method for action recognition, leveraging visual prompting techniques. The experimental results show the effectiveness of the proposed method.

**Strengths:**

The idea of prompt-offset tuning for few-shot class-incremental learning is interesting.

**Weaknesses:**

1. The motivation is not clear to me.  While the method has been tested for action tasks, notably skeletal action recognition, it doesn't seem tailored for this particular domain. Why was the method applied to action recognition and not to a standard few-shot class-incremental learning task?

2. The compared methods should be the few-shot class-incremental learning works, instead of the prompt tuning works. Drawing comparisons with prompt tuning methods, which aren't tailored for few-shot scenarios, might not provide a balanced perspective.

3. What is the meaning of * ** ^ in Table 3. How can the method achieve around 3% performance for both old and new classes?

4. To the best of my knowledge, few-shot class-incremental learning and continual few-shot learning are indeed distinct tasks. It might be worthwhile to revisit and clarify this in the study.

**Questions:**

Please refer to 'Weaknesses'